# Multiomics Signature Reveals Network Regulatory Mechanisms in a CRC Continuum

**DOI:** 10.3390/ijms26157077

**Published:** 2025-07-23

**Authors:** Juan Carlos Higareda-Almaraz, Francesco Mattia Mancuso, Pol Canal-Noguer, Kristi Kruusmaa, Arianna Bertossi

**Affiliations:** 1Universal Diagnostics S.A., 41013 Seville, Spain; jc.higareda@universaldx.com (J.C.H.-A.); francesco.mancuso@universaldx.com (F.M.M.); pol.canal@universaldx.com (P.C.-N.); 2Research & Development, Universal Diagnostics d.o.o., 1000 Ljubljana, Slovenia; kristi.kruusmaa@universaldx.com

**Keywords:** colorectal cancer, advanced adenomas, methylation, high-grade dysplasia, low-grade dysplasia, multiomics, regulatory network, early detection

## Abstract

Sporadic colorectal cancer (CRC), the third leading cause of cancer-related death globally, arises through a continuum from normal tissue to adenomas, progressing from low-grade (LGD) to high-grade dysplasia (HGD); yet, the early epigenetic drivers of this transition remain unclear. To investigate these events, we profiled LGD and HGD adenomas using EM-seq, and identified a consensus differential methylation signature (DMS) of 626 regions through two independent bioinformatics pipelines. This signature effectively distinguished LGD from HGD in both tissue and plasma-derived cell-free DNA (cfDNA), highlighting specific methylation patterns. Functional annotation indicated enrichment for regulatory elements associated with transcription factor activity and cell signaling. Applying the DMS to the TCGA CRC dataset revealed three tumor subtypes with increasing hypermethylation and one normal cluster. The most hypermethylated subtype exhibited poor survival, high mutation burden, and disrupted transcriptional networks. While overlapping with classical CpG Island Methylator Phenotype (CIMP) categories, the DMS captured a broader spectrum of methylation alterations. These findings suggest that the DMS captures functionally relevant, antecedent epigenetic alterations in CRC progression, enabling the robust stratification of dysplasia severity and tumor subtypes. This signature holds promise for enhancing preclinical detection and molecular classification, and warrants further evaluation in larger prospective cohorts.

## 1. Introduction

Colorectal cancer (CRC) is a leading cause of cancer-related mortality worldwide and poses a significant public health burden [1]. Despite advances in clinical management, current classification systems rely primarily on histopathological features and staging, which often fail to reflect the underlying molecular heterogeneity observed among patients [2]. High-throughput sequencing technologies have since revealed a complex landscape of genetic and epigenetic alterations that drive CRC progression and influence patient outcomes. Among these, DNA methylation has emerged as a critical epigenetic regulatory mechanism implicated in tumorigenesis [3]. However, the precise interplay between genome-wide methylation patterns, somatic mutation burden, and clinical trajectories remains incompletely understood [4].

Importantly, CRC does not arise de novo, but rather through a well-characterized adenoma–carcinoma sequence, in which, in colorectal adenomas, normal colonic epithelium progresses to low-grade dysplasia (LGD), then high-grade dysplasia (HGD), and ultimately to invasive carcinoma [5]. This stepwise model is underpinned by cumulative molecular changes, including both genetic mutations and epigenetic modifications. While global methylation alterations have been widely studied in established CRCs [6], the foundational methylation events marking the LGD–HGD transition remain underexplored, despite their potential for early risk stratification and non-invasive detection [7].

Large-scale consortia such as The Cancer Genome Atlas (TCGA) [8,9,10], the International Cancer Genome Consortium (ICGC) [11], the Catalog of Somatic Mutations In Cancer (COSMIC) [12], or the Cancer Dependency Map (DepMap) [13], provide an unprecedented opportunity to interrogate CRC biology across multiple molecular layers. By integrating DNA methylation profiles with gene expression, somatic mutations, and clinical outcomes, researchers can derive more nuanced molecular subtypes of cancer that better reflect the underlying biology [14]. However, even with these resources, patient outcomes remain highly variable, highlighting the need for a deeper integration of multiomic signals to refine prognostic and therapeutic frameworks [15,16].

Previous efforts to classify CRC at the molecular level have yielded important frameworks such as the CpG Island Methylator Phenotype (CIMP), which identifies a subset of tumors with widespread promoter hypermethylation [17], and the Consensus Molecular Subtypes (CMSs), which consolidate transcriptomic patterns into four biologically distinct groups [18]. While these classifications have enhanced our understanding of CRC diversity, both have limitations, particularly in capturing the full range of tumor heterogeneity, especially at early disease stages or within intermediate dysplastic lesions. This persistent molecular heterogeneity underscores the need for integrative approaches that leverage both epigenetic and genetic features across the adenoma–carcinoma continuum.

Although key genetic drivers such as *APC*, *KRAS*, and *TP53* are well established in CRC, these alone cannot fully explain disease heterogeneity [19,20]. Epigenetic alterations, particularly DNA methylation, add an additional regulatory dimension that may demarcate distinct tumor subtypes. Integrating methylation data with mutation burden and transcription factor (TF) networks offers the potential to uncover regulatory circuits that drive tumorigenesis and may be amenable to early intervention or targeted therapy [21,22,23,24]. Certain methylation patterns may correlate with genomic instability, offering opportunities to delineate specific pathways of tumorigenesis or identify new points of therapeutic intervention [25]. By incorporating stemness-related gene expression and TF network analysis, it becomes possible to elucidate the molecular switches orchestrating these methylation-defined subtypes, and identify candidate biomarkers for early detection and precision oncology [26,27].

In parallel, integrative multiomics approaches have become increasingly powerful tools for molecular classification [28], enabling patient stratification by combining genomic, epigenomic, and transcriptomic data. These methods often rely on high-dimensional representations [29] or latent variable models [30], to define the subtypes associated with clinical outcomes. Other common strategies include perturbation-based clustering [31], multi-kernel learning [32], deep learning frameworks [33], and matrix fusion techniques [34]. However, these frameworks are primarily data-driven, often designed for general-purpose clustering or dimensionality reduction, and typically lack explicit incorporation of biological knowledge, behaving as a “black box” [35]. Moreover, they are frequently trained on late-stage tumors, which may make pre-neoplastic molecular events critical for disease initiation and progression obscure. This limits their capacity to capture the biological underpinnings of early dysplastic transitions, particularly in the context of adenoma-to-carcinoma progression.

To address these gaps, the present study leverages comprehensive multi-platform data spanning advanced adenomas (AAs) and CRCs to explore the molecular continuum from LGD to HGD to invasive disease [36]. We aimed to identify epigenetically defined subtypes of CRC using a genome-wide DNA methylation signature derived from lesions at various dysplasia stages. By integrating these methylation patterns with somatic mutation burden and TF network activity, we investigated whether specific epigenetic profiles correlate with poor clinical outcomes and genomic instability. Additionally, we focused on uncovering key regulatory factors, including stemness-related genes and transcriptional hubs, that may govern the progression of these subtypes. Our integrated framework highlights early molecular events that may serve as biomarkers or therapeutic targets, and offers a foundation for translational applications such as non-invasive screening via cell-free DNA (cfDNA) and personalized treatment strategies.

## 2. Results

We conducted a comprehensive multiomics analysis using a three-step approach (Figure 1). In the first step, our differential methylation strategy identified 3125 differentially methylated tiling windows using *MethylKit*, and 31,314 differentially methylated bins using *DMRcaller*. From these, 626 overlapping regions were identified, collectively constituting a differential methylation signature (DMS). DMS displayed methylation patterns that effectively distinguished high-grade dysplasia (HGD) from low-grade dysplasia (LGD) in both tissue and plasma samples.

### 2.1. Methylation Differences Between HGD and LGD Stratifies Dysplasia in Tissue and Plasma

Differentially methylated regions (DMRs) between HGD and LGD showed distinct epigenetic landscapes associated with lesion severity (Figure 2a). HGD-hypermethylated DMRs were enriched in regulatory genomic regions, including enhancers, promoters, 5′ UTRs, exons, CpG islands, and shores. In contrast, HGD-hypomethylated regions were predominantly located in the intronic, upstream (1–5 kb), and intergenic regions (Figure 2b). This distribution pattern suggests that methylation changes target gene regulatory elements as dysplasia progresses.

Principal component analysis (PCA) based on the differential methylation signature (DMS) separated AA tissue samples by dysplasia grade, confirming the signature’s capacity to capture transformation-related epigenetic alterations (Figure 2c). A similar separation was observed in plasma-derived cfDNA AA samples, where the DMS-based analysis separated the LGD and HGD cases. Additionally, DMS distinguished AA plasma samples from healthy patient plasma and control buffy coat samples (Figure 2d). This indicates that cfDNA methylation profiles reflect the underlying tissue changes and hold potential as noninvasive markers for tracking dysplasia progression.

### 2.2. Functional Enrichment Analysis

Gene Ontology (GO) and pathway enrichment analyses revealed distinct biological processes enriched in HGD and LGD (Figure 3). Hypermethylated HGD DMRs were enriched in pathways associated with DNA-binding transcription factors, tissue morphogenesis, and the regulatory networks involved in cellular transformation (Figure 3a). On the other hand, hypomethylated DMRs revealed transcription factors associated with alternative mechanisms of early neoplastic progression (Figure 3b). These results further reinforce the notion that epigenetic alterations influence key biological processes underlying the progression of colorectal dysplasia.

### 2.3. Methylation-Based Tumor Stratification and Clinical Implications

Unsupervised hierarchical clustering of the TCGA COAD-READ dataset using the DMS signature identified four distinct clusters (CLs): CL1 (normal samples) and three unique tumor subtypes (CL2, CL3, and CL4) (Figure 4a). Analysis of mean methylation levels across clusters revealed a progressive, stepwise increase from CL1 to CL4, with CL2 and CL3 occupying intermediate states and CL4 exhibiting the highest global methylation levels (Figure 4b). PCA confirmed CL4 to be the most epigenetically divergent from normal tissue, whereas CL2 and CL3 displayed intermediate profiles (Figure 4c).

These findings illustrate a continuum of progressive hypermethylation across the clusters: CL4 exhibits the highest degree of hypermethylation, followed by CL3 and CL2 with intermediate levels, culminating in CL1 (normal), which shows the lowest methylation levels (basal methylation).

This methylation continuum is strongly correlated with clinical outcomes. Kaplan–Meier survival analysis revealed that CL4 correlated with a significantly poorer prognosis relative to CL2, indicating that extensive hypermethylation in CL4 is associated with more aggressive tumor phenotypes (Figure 4d).

Consistent with its poor survival outcomes, CL4 also exhibited the highest mutation burden, reinforcing the link between epigenetic dysregulation, genomic instability, and tumor aggressiveness (Figure 4e).

Interestingly, an inverse trend emerged in the EMTes analysis compared to the methylation pattern. Epithelial–mesenchymal transition-related genes (ITGB1, VIM, MMP2, COL1A1, TWIST1, CTNNB1, FN1, SNAI2, ITGB6, CDH1, CDH2, MMP9, ZEB1, and FOXC2) showed the highest mean Z-score expression in CL3, moderate levels in CL2, and generally lower expression in CL4 relative to the other tumor clusters (Figure 4f).

Together, these observations underscore the prognostic value of DMS-driven clustering in colorectal cancer, offering critical insights into potential subtype stratification, invasion mechanisms, and plausible targets for therapeutic strategies.

### 2.4. Network-Level Insights into Transcription Factors Highlight Oncogenic and Immune Divergence Across CRC Subtypes

To elucidate the regulatory mechanisms underlying these methylation-based clusters, we performed a transcription factor (TF) network analysis. Network topology revealed distinct TF hubs and bottleneck regulators for each cluster, highlighting the unique oncogenic and immune-related pathways across CRC subtypes. In CL4, which was associated with the poorest survival, USF2, TWIST1, ZNF143, and LYL1 emerged as hubs with a positive regulation of downstream targets and high bottleneck/betweenness, whereas ZBTB7A acted as a hub with a negative regulation of downstream targets and similarly high network centrality (Figure 5a). Functionally, CL4 was enriched in canonical cancer pathways, including hepatocellular carcinoma, gastric cancer, small cell lung, prostate, and bladder cancers, as well as in key signaling pathways such as PI3K–Akt and AMPK (Figure 5b). Gene Ontology (GO) analysis further underscored CL4’s involvement in the reactivation of developmental pathways and the regulation of cell adhesion (Figure 5c).

In CL2, the TF network identified ELF5 and ZKSCAN1 (Figure 5d) as hubs with positively regulated downstream targets, whereas PAX8, CEBPB, ZNF639, and STAT6 functioned as hubs with negatively regulated downstream targets. KEGG pathway analysis identified immuno-inflammatory signaling pathways, such as JAK–STAT, TNF, IL-17, T-cell receptor, and NF-κB (Figure 5e), while GO terms revealed enrichment for leukocyte adhesion, proliferation, and migration (Figure 5f), suggesting a pronounced immunomodulatory profile.

Finally, the CL3 TF network displayed hubs with positive downstream regulation, including KLF5, TCF4, ZBED1, ZKSCAN1, and XBP1, along with the notable non-hub ZNF384 (Figure 5g), which showed a strong downstream target expression. In contrast, ZEB2, SPI1, and CREB1 have emerged as hubs that negatively regulate their downstream targets. The KEGG enrichment in CL3 reflected a heterogeneous landscape, encompassing PI3K–Akt signaling, transcriptional misregulation in cancer, cell lineage differentiation, and inflammatory pathways such as AGE–RAGE and the intestinal immune network (Figure 5h). Biological processes were similarly diverse, with enrichment in cell adhesion, leukocyte proliferation and activation, and the regulation of immune effectors (Figure 5i), collectively suggesting a distinctive immuno-oncological phenotype among tumor clusters.

Taken together, these results indicate a complex interplay between epigenetic modifications and TF-driven regulatory networks that define distinct oncogenic and immune phenotypes across CRC subtypes. This multifaceted regulation underscores the potential of integrating methylation data, network analyses, and pathway insights to refine the prognostic stratification and guide more precise therapeutic interventions.

### 2.5. CIMP Stratification Reveals Epigenetic Convergence with DMS-Based Clusters

To assess whether our DMS-based methylation clusters aligned with established epigenetic classifications of CRC, we first evaluated the CpG Island Methylator Phenotype (CIMP) status across TCGA COAD-READ samples. The CIMP status was derived using two canonical classification panels, Weisenberger and Ogino, based on gene-level hypermethylation relative to normal reference tissues.

In the tumor-only samples, the Weisenberger panel identified 37 (9.0%) cases as CIMP-high (CIMP-H), 111 (27.1%) as CIMP-low (CIMP-L), and 262 (63.9%) as CIMP-negative (CIMP-N). The Ogino panel, which included three additional loci, classified 31 (7.6%) samples as CIMP-H, 74 (18.0%) as CIMP-L, and 305 (74.4%) as CIMP-N. When including normal samples (labeled as CIMP-N by definition), the total sample counts increased to 455, with minimal shifts in the proportional distribution (Weisenberger: 307 CIMP-N; Ogino: 350 CIMP-N).

The integration of CIMP labels into the hierarchical clustering of DMS-defined differentially methylated regions (DMRs) revealed strong enrichment of CIMP-H samples within cluster CL4, the subgroup displaying the most extensive hypermethylation (Figure 6). Both classification panels consistently mapped CIMP-H cases predominantly to CL4, whereas clusters CL2 and CL3 largely comprised CIMP-N and CIMP-L samples. The normal samples in cluster CL2 uniformly corresponded to CIMP-N, confirming the baseline methylation status.

Notably, CL4, the most epigenetically aberrant cluster, coincided with poor clinical outcomes, highest mutation burden, and transcriptional signatures enriched in oncogenic pathways. These observations support the biological coherence of CL4 with the canonical CIMP-H phenotypes. However, the limited number of samples classified as CIMP-H by either panel (≤9%) contrasts with the broader epigenetic landscape captured by DMS. This suggests that while traditional CIMP panels identify a subset of highly methylated tumors, DMS encompasses a more comprehensive continuum of methylation dysregulation that also stratifies intermediate phenotypes such as CL2 and CL3.

Collectively, these findings underscore the ability of DMS to capture clinically relevant methylation patterns. While overlapping with the established CIMP-H category, DMS-based clustering probably revealed greater resolution across the CRC epigenomic spectrum, including early-stage and intermediate methylation profiles that may not be fully captured by traditional CIMP criteria.

## 3. Discussion

In this study, we present a biology-driven, hypothesis-guided strategy to identify early DNA methylation alterations in colorectal adenomas and explore their relevance across the CRC continuum. By profiling methylation patterns in LGD and HGD lesions with increasing potential to progress to CRC, we defined a DMS that captures early epigenetic disruptions. We then integrated this signature with multiomics data and biological network models to assess its implications in advanced CRC, including tumor subtyping, regulatory remodeling, and patient outcomes. Importantly, we also showed the feasibility of detecting this signature in cfDNA, highlighting its potential for noninvasive early detection.

Unlike recent computational strategies that use some variation of unsupervised clustering [31,32,34] or deep learning [33,37] to define subtypes de novo from late-stage, large-scale datasets such as TCGA, our approach incorporates biological priors derived from early lesions. These data-driven models have proven useful for tumor stratification but are typically applied to established cancers and often overlook initiating molecular events. Our framework complements such methods by anchoring subtype discovery in precancerous tissues, allowing us to examine epigenetic and regulatory shifts throughout disease progression, and enhance our understanding of CRC evolution from its earliest detectable stages.

Other recent studies have attempted to incorporate biological knowledge into subtype modeling using systems biology and machine learning. For example, Nakazawa and collaborators [38] constructed Bayesian transcriptional networks for personalized classification across multiple cancer types, while Bai et al. [39] developed a multiomics pipeline focused on programmed cell death in gastric cancer. These studies represent important steps toward personalized classification, but they are often limited by their focus on transcriptomic data and late-stage tumors, with less integration of early events or mechanistic insight. In contrast, our study emphasizes biological interpretability by integrating early-stage methylation changes with transcriptional regulatory context, mapping these changes along the adenoma–carcinoma continuum to link them to phenotypes and outcomes.

### 3.1. Early Epigenetic Dysregulation in Adenomatous Lesions

Our data show that HGD lesions exhibit distinct DNA methylation alterations in regulatory regions, which may precede canonical driver mutations in APC, KRAS, and TP53. Functional enrichment revealed that these early epigenetic changes involve genes related to tissue architecture, morphogenesis, and cell signaling, i.e., the pathways implicated in tumor initiation.

These results are supported by prior work. Luo and collaborators [40] profiled genome-wide methylation patterns across normal mucosa, adenomas, and CRCs, identifying high- and low-frequency methylator adenomas. They observed that certain adenomas shared methylation profiles with cancers, including intergenic and intragenic CpG changes and driver mutations in KRAS and APC, suggesting that epigenetic reprogramming begins early and persists through progression. Additional genome-wide studies on LGD and HGD samples have similarly identified hundreds of differentially methylated CpG sites compared to normal tissue, with many alterations maintained in CRC [7,41].

### 3.2. cfDNA Reflects Tumor-Specific Methylation Changes

We further demonstrated that the DMS could stratify cfDNA samples from patients with LGD and HGD, providing evidence that these early methylation signals can be detected noninvasively. This aligns with previous reports showing that cfDNA methylation profiling can identify tumor-associated signatures, even at preclinical stages [29,42,43].

Although our cfDNA cohort is relatively small, these findings contribute to a growing body of evidence supporting methylation-based liquid biopsies as early detection tools and vehicles for risk stratification. Our ability to distinguish LGD from HGD using cfDNA suggests that methylation profiling may guide surveillance and intervention strategies, pending validation against established biomarkers. Unlike stool-based or endoscopic screening methods, which may miss early molecular transitions and are invasive [44,45,46], cfDNA methylation assays may capture dysplasia earlier and more efficiently. This is consistent with broader studies suggesting that liquid biopsies can reveal tumor heterogeneity and inform clinical decisions [47,48].

### 3.3. DMS Reveals a Continuum and a Branching of Epigenetic States

The application of the DMS to TCGA COAD/READ datasets revealed four tumor clusters (CL1–CL4), spanning a methylation gradient from normal (CL1) to hypermethylated (CL4) states. Importantly, intermediate clusters (CL2 and CL3) displayed similar methylation levels but diverged in transcription factor activity, EMT features, and clinical outcome.

CL2 was enriched for immune- and inflammation-related TFs (STAT6, CEBPB), exhibited reduced EMT signaling, and was associated with favorable survival. CL3, by contrast, showed the activation of EMT-related TFs (ZEB2, SPI1), more invasive traits, and poorer outcomes, despite similar mutation burdens. These findings suggest that early epigenetic alterations may trigger distinct tumor trajectories, one involving immune modulation and better prognosis, another marked by plasticity and aggressiveness.

This branching is consistent with previous CRC classifications [49,50], including the CMS system, which defines four subtypes ranging from immune-enriched MSI (CMS1) to mesenchymal tumors (CMS4). While CMS subtypes are based on late-stage transcriptomic data and are influenced by stromal/immune composition, our methylation-based approach may capture earlier, tumor-intrinsic differences that precede transcriptional divergence.

Importantly, we recognize that epigenetic dysregulation and tumor progression involve complex feedback loops. The relationship is not strictly unidirectional [51,52]. Studies have shown that mutations in TP53 [53] and KRAS [54] can actively reshape the epigenome [55,56,57]. Conversely, epigenetic priming may also precede mutation acquisition [9]. Our findings support a model in which early methylation changes guide certain tumor trajectories, while remaining susceptible to modulation by co-occurring genetic events.

### 3.4. Transcription Factor Network Rewiring

Our TF network analysis further supports the epigenetic branching model. CL4 tumors are dominated by oncogenic TFs such as TWIST1 and USF2, linked to high methylation, poor outcomes, and aggressive phenotypes. CL2 exhibits immune-related TF activity (STAT6, CEBPB), while CL3 shows elevated EMT drivers (ZEB2, SPI1), consistent with its invasive behavior.

These patterns reflect the concept of TF network plasticity, in which transcriptional circuitry is dynamically rewired during malignancy. This idea was formalized by Hnisz [58] and is now recognized as a hallmark of cancer, with Hanahan [55] describing epigenetic reprogramming and enhancer remodeling as key to cellular identity shifts. Super-enhancer–driven core transcriptional regulatory circuits (CRCs) have also been described in leukemia, neuroblastoma, and other malignancies [59,60].

Together, our findings suggest that methylation-driven changes may seed the formation of distinct TF circuits, shaping tumor identity, plasticity, and vulnerability. This highlights the value of integrating TF network analysis into methylation-based subtyping strategies (Figure 7).

### 3.5. DMS vs. Classical CIMP Classification

Although CL4 overlaps with the canonical CIMP-high phenotype, nearly half of its tumors were classified as CIMP-low by established panels (e.g., Weisenberger, Ogino). This illustrates the limitations of threshold-based CIMP classification, which may not fully capture functional methylation heterogeneity, particularly in regulatory regions.

Rather than replacing CIMP, our model adds resolution by contextualizing methylation within TF and enhancer networks. Emerging classifications such as the CpG-enhancer methylator phenotype (CEMP) [61] and TF–CpG–gene regulatory modules [62,63] support this integrated view. By linking methylation to regulatory dynamics, our framework offers a mechanistically grounded alternative to fixed CpG panels.

### 3.6. Translational Outlook and Study Limitations

Early methylation changes, detectable in both tissue and cfDNA, may enable the noninvasive detection and molecular stratification of CRC. The ability of the DMS to distinguish LGD from HGD in cfDNA supports its clinical utility for identifying high-risk precursors. Furthermore, the epigenetic clustering of tumors reveals functionally distinct subtypes, each with different regulatory programs, immune profiles, and clinical trajectories.

However, several limitations should be acknowledged. First, the sample size, particularly for cfDNA analyses, was relatively small. Validation in larger, multi-center cohorts will be essential to assess the reproducibility and clinical utility of these findings. Second, the mechanistic inferences are based primarily on correlative analyses. Functional validation of transcription factor activity and direct causal links between methylation changes and phenotypic outcomes are necessary to support our interpretations. Third, while our approach captures early regulatory changes, its performance must be benchmarked against current clinical workflows and evaluated across more diverse patient populations. Finally, the specificity of the DMS for CRC remains to be established through direct comparisons with other malignancies and benign gastrointestinal conditions.

Despite these limitations, our integrative, biology-driven approach provides a strong conceptual and methodological foundation for future studies aiming to refine CRC classification and enable early detection using methylation and regulatory network data.

## 4. Materials and Methods

### 4.1. Sample Preparation and Whole-Genome EM-seq

A total of 73 AA tissue samples were included in this study. Fifteen fresh-frozen (FF) specimens were obtained from the Indivumed Biobank repository (Hamburg, Germany), while 58 blood samples were collected as part of a prospective multicenter study conducted in Spain, Germany, and Ukraine.

For the initial differential methylation analysis, nine HGD and six LGD AA tissue samples were selected (Table 1). Subsequently, for cfDNA analysis, a separate cohort of 30 AA cases (26 LGD and 4 HGD) and 28 control samples (10 healthy plasma and 18 buffy coat samples) were examined to verify whether the identified tissue-specific differential methylation was also detectable in cfDNA and remained tumor-specific (Table 2). The detailed sample characteristics are presented in Appendix A.

Genomic DNA was extracted from buffy coat and FF tissue using the Qiagen DNeasy Blood and Tissue Kit (Qiagen, Valencia, CA, USA), following the manufacturer’s protocol. Cell-free DNA (cfDNA) extraction from plasma was performed using the Mag-Bind cfDNA Kit (Omega Bio-Tek, Norcross, GA, USA) according to the manufacturer’s instructions. Subsequently, DNA from each sample was used for library preparation, with artificial spike-in controls from the Premium RRBS kit V2 (Diagenode, Belgium), which was used to monitor enzymatic conversion efficiency following the manufacturer’s instructions.

Library preparation and conversion were carried out using the NEBNext Enzymatic Methyl-seq (EM-seq) Kit (New England Biolabs, Ipswich, MA, USA). EM-seq adaptors were ligated to the processed DNA, and enzymatic conversion was performed using TET2 and an oxidation enhancer, followed by APOBEC-mediated deamination. After conversion, indexing and amplification PCR were conducted using NEBNext Q5U Master Mix (New England Biolabs, Ipswich, MA, USA).

The prepared libraries were sequenced using a NovaSeq 6000 (Illumina, San Diego, CA, USA) with paired-end sequencing (2 × 150 bp) to an average depth of 24×.

### 4.2. Sequence QC and DNA Methylation Analysis

Quality checks and trimming were performed using *FastQC* (v0.12.1) [64] *TrimGalore* (v0.4.5) (a wrapper tool around *Cutadapt* [65], which removed adapter sequences and poor-quality bases and reads).

The remaining high-quality reads (average Phred score > 35) (Appendix A) were aligned to a bisulfite-converted human genome (Ensembl 91 assembly, hg38) using the *Bismark Bisulfite Read Mapper* (v0.20.0) [66].

Methylation calls for every C analyzed were performed using the *Bismark methylation_extractor* script. For each CpG, the beta values (β) were calculated as follows:β = CGmethylated/(CGmethylated + CGunmethylated)
where CGmethylated is the number of methylated cytosines, and (CGmethylated + CGunmethylated) is the sum of methylated and unmethylated cytosines (total number of reads) at that position.

Differential methylation analysis (DMA) was conducted in two sequential steps to ensure robustness and accuracy. First, differentially methylated regions (DMRs) were identified using two independent bioinformatics pipelines: *DMRcaller* (v1.40.0) [67] and *MethylKit* (v1.34.0) [68] R packages. Bismark cytosine reports aligned to the UCSC hg38 reference genome from HGD and LGD tissue samples were used for analysis.

For *DMRcaller*, parameters were set as follows: context = “GC”, proportion = “true”, *p*-value threshold = 0.01, minimum cytosine count = 4, and minimum reads per cytosine = 4. Genomic bins of 100 bp with a 100 bp step size were generated, and methylation levels were summarized within these bins.

For *MethylKit*, the context was set to “CpG”, with a minimum coverage of 4. Tiling windows of 1000 bp with a 1000 bp step size were generated, and methylation levels were summarized within these tiles. A logistic regression model was applied for differential analysis, using a q-value cutoff of 0.01 and a methylation cutoff of 25%, incorporating biological sex as a covariate.

To enhance the biological relevance and minimize algorithmic artifacts, the intersection of the results from both pipelines was selected, defining these regions as the differentially methylated signature (DMS).

Principal component analysis (PCA) was performed using beta values from differentially methylated regions (DMRs) to visualize methylation cluster separation in a reduced-dimensional space. Analysis was conducted using the *FactoMineR* (v2.11) package [69]. The samples were color-coded according to their cluster assignment, and confidence ellipses were added to highlight the cluster boundaries.

To evaluate potential confounding effects of geographic origin, we performed a batch-wise comparison of global methylation levels across cfDNA samples from Spain, Germany, and Ukraine. No statistically significant differences were detected (ANOVA, *p* > 0.05), supporting the robustness of the methylation signal and its association with lesion status rather than sample geographical origin.

### 4.3. Integration of TCGA Multimodal Data and Analysis

To extend the analysis to The Cancer Genome Atlas (TCGA) colorectal adenocarcinoma (COAD) and rectum adenocarcinoma (READ) datasets [70], a bin-based approach was used to summarize the DNA methylation data. Methylation probes from TCGA were aggregated into genomic bins based on their genomic coordinates using the UCSC hg38 genome built using *TCGAbiolinks* (v2.36.0) [71]. The mean methylation levels were calculated for all CpG probes within each bin. Bins lacking valid methylation data were excluded from analysis. This preprocessing step produced a bin-level methylation matrix, enabling integration with additional data types, such as gene expression and mutation data.

Gene expression data were linked to methylation bins by identifying genes within 5 kb of each bin using the UCSC-known gene database. The gene identifiers were converted into gene symbols to ensure consistency. The somatic mutation data were processed to create a binary mutation matrix for the 20 most frequently mutated genes in the dataset. Samples were classified as mutated (1) or wild-type (0) for each gene, and sample identifiers were standardized across the methylation, gene expression, and mutation datasets.

Differential methylation between TCGA samples was performed at the bin level. Bins were classified based on their methylation differences between LGA and HGA, based on the original dysplasia signature. Specifically, bins were labeled as hypermethylated in HGA if the methylation difference was more than 20%, hypomethylated if the difference was less than 20%, or unchanged. Bins were linked to nearby genes, facilitating downstream functional analyses.

The final bin-level methylation matrix underwent *ConsensusClusterPlus* (v1.72.0) R package [72] to determine the optimal number of clusters (k). The analysis was performed on z-score-normalized methylation data, with the following parameters: maximum number of clusters (maxK) set from 3 to 20, 1000 resampling iterations (reps), 80% item sampling proportion (pItem), 100% feature sampling proportion (pFeature), hierarchical clustering algorithm (clusterAlg = “hc”), and Euclidean distance metric (distance = “euclidean”). A random seed was used to ensure reproducibility. Final clustering results were visualized using the *ComplexHeatmap* (v2.24.1) package [73], allowing for the identification of both sample-level and genomic region–level cluster structures.

### 4.4. Clinical Analysis and Epithelial–Mesenchymal Transition Expression Signature (EMTes)

The mutation burden, calculated as the total number of mutations per sample, was analyzed across the methylation-based clusters. Statistical comparisons were performed using the Kruskal–Wallis test for overall differences and post hoc Wilcoxon rank-sum tests with Benjamini–Hochberg corrections for pairwise comparisons (*p* < 0.05).

Kaplan–Meier survival analysis was performed to evaluate patient survival outcomes. Log-rank tests using the R package *Survminer* (v0.5.0) [74] were utilized to compare survival distributions among different methylation clusters, excluding normal samples (CL1). The analysis was restricted to primary tumor samples, and survival times were truncated at 1000 days, focusing on near-term prognosis, with right-censoring applied for patients still alive at this limit.

Pairwise survival comparisons were performed between selected clusters (CL2 vs. CL3, CL2 vs. CL4, and CL3 vs. CL4) to assess the differences in survival distributions. Kaplan–Meier plots with log-rank *p*-values, confidence intervals, and risk tables were generated to visualize survival trends between clusters.

Additionally, an overall survival analysis was performed across all tumor-associated clusters, with survival curves estimated using the Kaplan–Meier method. This allowed for a clear comparison of the survival probabilities among the different methylation-based groups.

The EMTes panel comprised 19 genes with a well-established participation in epithelial-to-mesenchymal transitions (ITGB1, VIM, MMP2, COL1A1, TWIST1, CTNNB1, SNAI1, FN1, SNAI2, TGFB1, ITGB6, SOX10, CDH1, CDH2, MMP9, TWIST2, ZEB1, ZEB2, and FOXC2). These genes have been consistently implicated in processes that underlie EMT, such as cell adhesion, extracellular matrix remodeling, and transcriptional regulation [75]. Transcriptome data from TCGA were first log-transformed (if not already in log scale). For each gene across all the samples in the dataset, we computed the mean and standard deviation of its expression. Each sample’s expression value for a given gene was then converted to a z-score as follows:z = (xi−)/σ μ
where xi is the expression value for the gene in the ith sample and μ and σ are the mean and standard deviation for that gene across the entire cohort, respectively. These z-scores were used to calculate the EMT score per sample by averaging the z-scores of the EMTes genes.

Subsequently, EMT scores were compared across methylation-based clusters using the non-parametric Kruskal–Wallis test to assess global differences. Pairwise comparisons between clusters were performed using the Wilcoxon rank-sum test. Unless otherwise noted, *p*-values < 0.05 were considered statistically significant for all analyses.

### 4.5. Transcription Factor Network and Functional Enrichment Analysis

Transcription factor (TF) activity was inferred using *DoRothEA* (v1.20.0) [76] regulons and *VIPER* (v1.42.0) [77] R packages, based on matching regulon target genes expressed per cluster. Networks were constructed based on correlation thresholds (0.4) and centrality metrics (degree, betweenness, and closeness) were computed to identify hub TFs. The results were visualized using *Cytoscape* (v3.10.3) [78] to highlight cluster-specific TF activity. All transcription factors identified through this analytical pipeline are supported by biologically validated interactions, as established in experimental studies conducted in diverse biological contexts [79,80,81,82].

Nearby genes linked to differentially methylated bins were subjected to enrichment analyses. Gene Ontology (GO) terms and KEGG pathways were identified using *enrichGO* and *enrichKEGG* functions, with Benjamini–Hochberg adjustments for multiple comparisons (*p*-value < 0.05). Gene Set Enrichment Analysis (GSEA) was conducted using cluster-associated genes to explore functional relationships. All analyses were performed using the *clusterProfiler* (v4.16.0) package [83]. All statistical analyses and data visualizations were performed using R (v4.5.1) and Bioconductor (v3.21).

### 4.6. CIMP Phenotype Calculation

To assess CpG Island Methylator Phenotype (CIMP) status, we utilized the complete COAD/READ-TCGA DNA methylation dataset, incorporating β-values derived from 455 CRC and normal tissue samples. Each genomic region was characterized by CpG site coverage and magnitude of methylation differences relative to normal tissues.

The CIMP status was determined using two widely accepted panels. The Weisenberger panel included five genes (CACNA1G, IGF2, NEUROG1, RUNX3, and SOCS1) [84], while the Ogino panel expanded this set with three additional genes (CDKN2A, MLH1, and CRABP1) [85]. Gene-level hypermethylation was defined by applying a Z-score transformation to the β-values relative to the normal tissue reference methylation levels on the associated probes. A gene was considered methylated if at least one-third of its associated probes exceeded the Z-score threshold of ≥3.

Subsequently, samples were classified into CIMP subtypes based on the number of hypermethylated genes. For the Weisenberger method, CIMP-high (CIMP-H) was defined as of more or equal to 3 methylated genes, CIMP-low (CIMP-L) as 2 methylated genes, and non-CIMP (CIMP-N) if 2 or less. Using the Ogino criteria, CIMP-H was defined as ≥5 methylated genes, CIMP-L as 3–4, and CIMP-N as <3.

Concurrently, we performed regional methylation analysis to identify differentially methylated regions (DMRs) between the tumor and normal samples. Regions exhibiting a mean β-value difference of >20% were designated as hypermethylated (gain), whereas those with <−20% were considered hypomethylated (loss). Regions with differences within ±20% were classified as unchanged. To ensure data quality, regions and samples with >50% of missing values were excluded. The remaining methylation matrix was z-score-normalized across regions (row-wise), enabling hierarchical clustering using Euclidean distance and Ward’s linkage. This approach facilitates the delineation of distinct methylation-driven subgroups.

Finally, the CIMP status and other relevant annotations were integrated into a heatmap visualization for comparative analysis, as described previously.

## Figures and Tables

**Figure 1 ijms-26-07077-f001:**
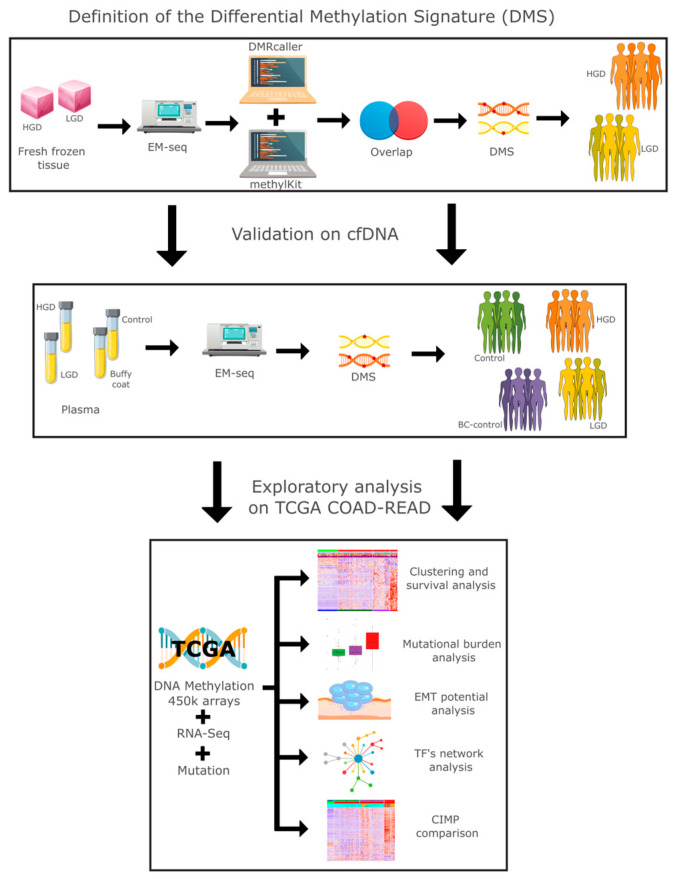
Study overview: graphical representation of the methodology from this study.

**Figure 2 ijms-26-07077-f002:**
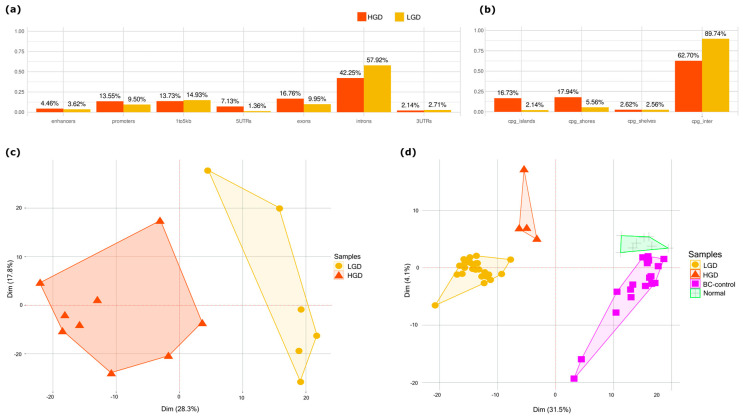
The distribution of differentially methylated regions between HGD and LGD. (**a**) Percentage of differentially methylated regions across genomic features (promoters, exons, introns, upstream, intergenic). (**b**) Percentage of differentially methylated regions across CpG contexts (CpG islands, shores, and intergenic CpG regions). (**c**) PCA analysis of tissue samples showing clear separation between HGD (*n* = 9) and LGD (*n* = 6) groups. (**d**) PCA analysis of plasma samples demonstrating the distinction between HGD (*n* = 4), LGD (*n* = 26), normal plasma (*n* = 8), and BC control (*n* = 18).

**Figure 3 ijms-26-07077-f003:**
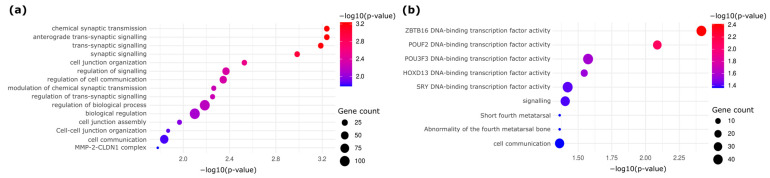
Functional enrichment analysis of DMS-associated regions. (**a**) GO analysis in HGD-enriched genes highlights the pathways involved in transcription factor regulation, tissue organization, and cell communication. (**b**) LGD-enriched genes are involved in different transcriptional and epigenetic regulatory mechanisms.

**Figure 4 ijms-26-07077-f004:**
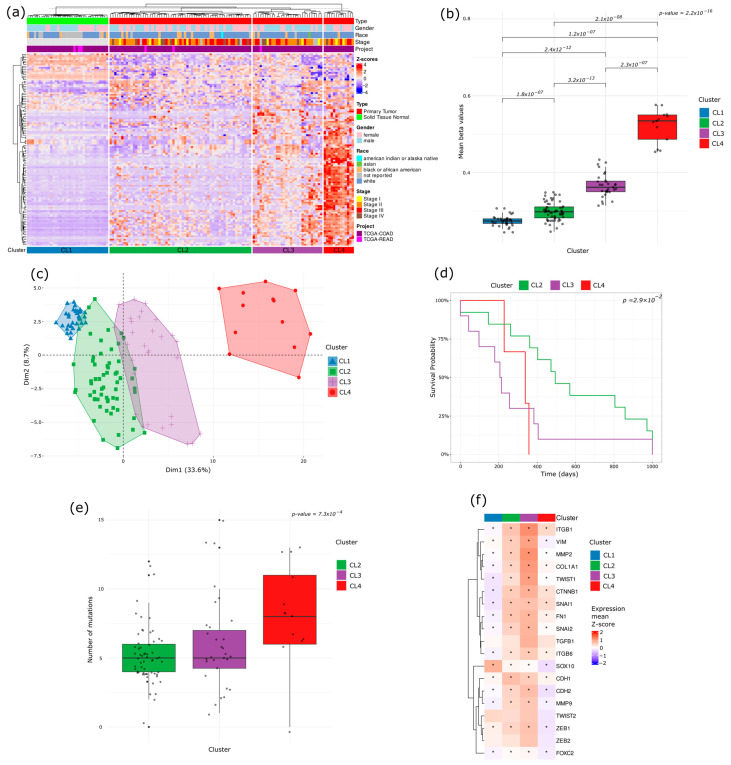
Clustering analysis of TCGA COAD-READ samples based on DMS regions. (**a**) Hierarchical unsupervised clustering identifies four sample clusters: CL1 (normal, blue) and three distinct tumor subtypes (CL2: green, CL3: purple, CL4: red). (**b**) Mean methylation levels per cluster reveal a stepwise increase from CL1 to CL4. (**c**) PCA analysis confirms CL4 as the most epigenetically divergent from normal tissue, while CL2 and CL3 share intermediate profiles. Clinical outcomes associated with methylation-based clusters. (**d**) Kaplan–Meier survival analysis shows a significantly lower survival probability in CL4 patients compared to CL2. (**e**) Mutation burden analysis reveals that CL4 has the highest mutation rate, supporting its aggressive nature. (**f**) EMT-related genes exhibit an inverse trend to methylation, with CL3 showing the highest expression, while CL4 displays significant downregulation. * indicate statistically significant differences determined by the Wilcoxon rank-sum test (*p* < 0.05).

**Figure 5 ijms-26-07077-f005:**
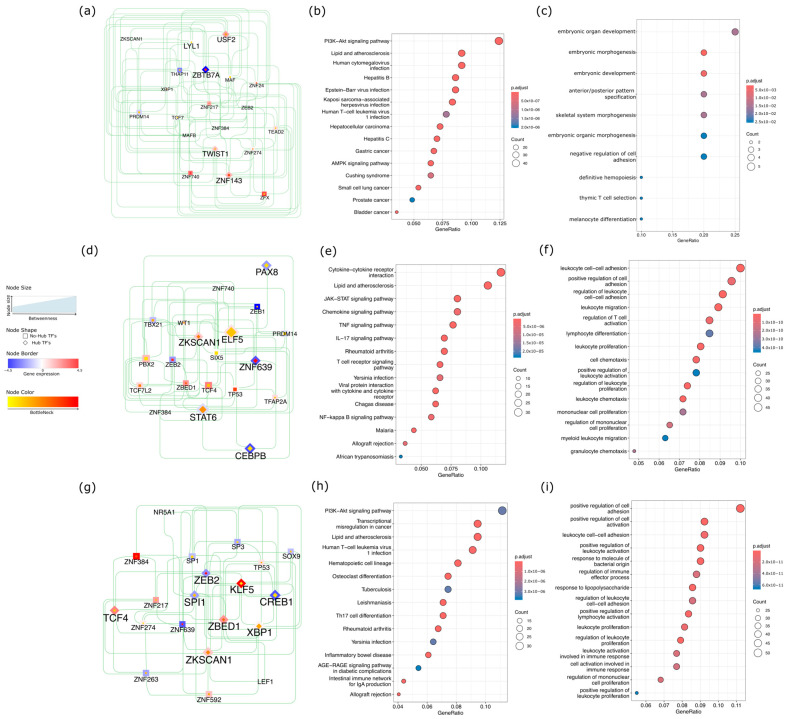
Transcription factor network and pathway enrichment analysis. (**a**) CL4 TF network featuring key transcriptional regulators and their centrality in driving oncogenic pathways; (**b**) KEGG pathway enrichment for CL4, highlighting canonical cancer pathways such as PI3K–Akt and AMPK; (**c**) Gene Ontology (GO) terms associated with CL4, illustrating embryonic gene programs and negative regulation of cell adhesion; (**d**) CL2 TF network depicting influential regulators of immune and inflammatory signaling; (**e**) KEGG pathway enrichment in CL2, emphasizing immuno-inflammatory pathways (JAK–STAT, TNF, IL-17, NF-κB); (**f**) GO terms for CL2, underscoring immune related terms as leukocyte adhesion, proliferation, and migration; (**g**) CL3 TF network revealing hubs shaping both oncogenic and immune-associated processes; (**h**) KEGG pathway enrichment in CL3, showing misregulation in cancer and PI3K–Akt, and heavily influenced on inflammatory pathways; (**i**) GO terms linked to CL3, reflecting the regulation of cell adhesion, leukocyte proliferation, and immune effector function.

**Figure 6 ijms-26-07077-f006:**
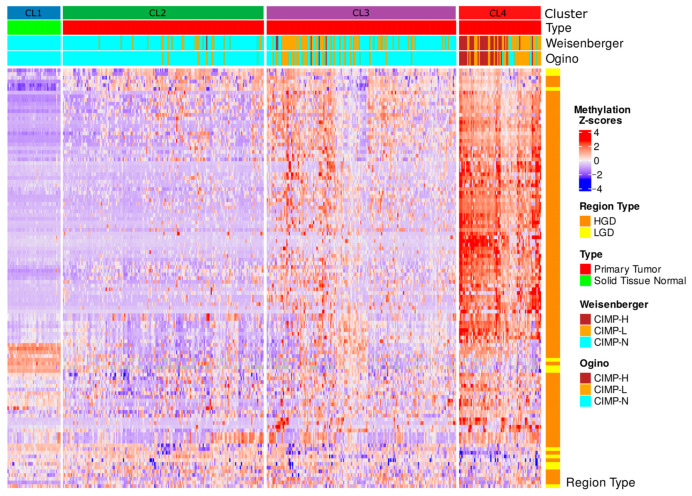
Integrative heatmap of methylation-based clustering and CIMP classification in TCGA COAD-READ samples. Heatmap displaying the hierarchical clustering of TCGA COAD-READ samples based on the DMS regions. Columns represent individual samples, grouped into four methylation-based clusters (CL1, CL2, CL3, CL4) according to hierarchical clustering. Rows correspond to DMS regions, with methylation levels represented as Z-scores. Top annotations indicate sample type and CIMP status determined by Weisenberger and Ogino panels (classified as CIMP-H, CIMP-L, CIMP-N). Right-side annotations classify DMS regions according to their association with HGD or LGD.

**Figure 7 ijms-26-07077-f007:**
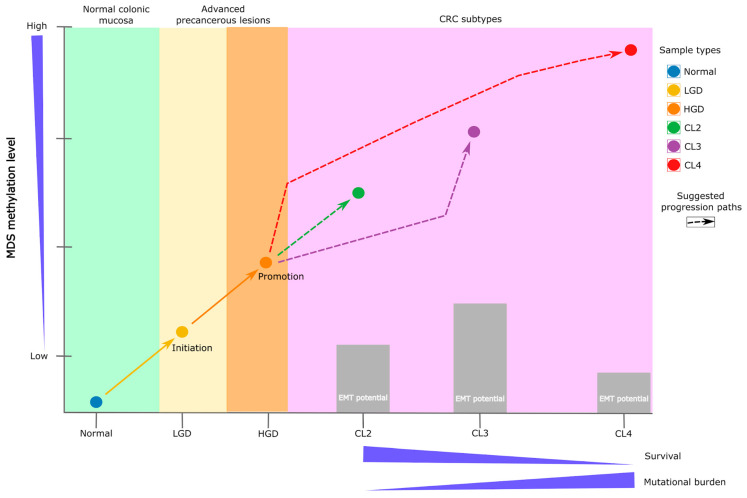
Schematic representation of CRC subtype progression. The initiation phase illustrates the transition from normal colonic mucosa to adenoma, marking the onset of the adenoma–carcinoma sequence. The promotion phase corresponds to dysplasia progression, during which specific epigenetic alterations arise. These changes are captured by the Differential Methylation Signature (DMS), which effectively distinguishes low-grade dysplasia (LGD) from high-grade dysplasia (HGD). Early methylation alterations precede malignant transformation and contribute to the development of distinct CRC subtypes. For instance, tumors classified as subtype CL4 are characterized by extensive hypermethylation, a higher mutational burden, and significantly poorer survival outcomes. In contrast, intermediate subtypes such as CL2 display moderate methylation levels, a lower mutation burden, and comparatively favorable prognoses.

**Table 1 ijms-26-07077-t001:** Tissue sample summary.

Diagnosis	*n*	Age ± SD	Sex Distribution	Age Range
HGD	9	67 ± 8	F 11%, M 88.9%	53–77
LGD	6	72.8 ± 9	F 66.7%, M 33.7%	59–82

**Table 2 ijms-26-07077-t002:** Plasma sample summary.

Diagnosis	*n*	Age ± SD	Sex Distribution	Age Range
HGD	4	63 ± 10.3	F 25%, M 75%	51–73
LGD	26	62.2 ± 6.5	F 53.8%, M 46.2%	51–74
Control	28	62.3 ± 12.2	F 53.6%, M 46.2%	30–84

## Data Availability

The data presented in this study are available upon reasonable request from the corresponding author.

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
