# Peer review of "Multiomics Signature Reveals Network Regulatory Mechanisms in a CRC Continuum"

_ijms, 2025, doi:10.3390/ijms26157077_

Round 1
Reviewer 1 Report
Comments and Suggestions for Authors
The authors of this manuscript aim to improve early detection and molecular classification of colorectal cancer. I consider this to be a topic of high interest for the patients since CRC has very invasive tests and very frequently delayed diagnosis. Below I state point by point my concerns:
- Could the authors state the year and the code of the ethics committee approbation. Especially for different countries that are involved.
- In the materials and methods section, could the authors provide a summarized description of their patients. Perhaps as a one table for blood samples and another table for tissue samples, since the groups are different and state known data (like age +SD, sex, histopathological diagnosis, or any other relevant data) .
- The authors have blood samples from different countries (Spain, Germany, and Ukraine). Did they check is there any significant difference between these groups of samples since the culture, geographical region and habits could influence different parameters like methylation.
- Since the idea of this work was to help early detection of CRC, could the authors explain how do they envision using their results in clinical sense? Did they plan to use patterns and compare them? How could they be sure that specific pattern is typical for CRC and not some other disease of condition, since they don’t have any other comparison?
- The discussion section is written as a mix of results with conclusions. The authors don’t discuss almost any of the works by other authors. Could the authors include more discussions into this section. Perhaps citing other works that have the same results supporting their own or other works that have opposite results and explaining why could that be.
Author Response
We sincerely thank Reviewer 1 for the thoughtful and constructive feedback on our manuscript. Your comments have been instrumental in refining both the scientific content and presentation of our work. Below, we provide detailed responses to each point raised, along with the corresponding changes made in the manuscript. All modifications have been clearly marked in the revised version. Please do not hesitate to let us know if any additional clarifications are needed.
Comment 1: The authors of this manuscript aim to improve early detection and molecular classification of colorectal cancer. I consider this to be a topic of high interest for the patients since CRC has very invasive tests and very frequently delayed diagnosis. Below I state point by point my concerns:
Could the authors state the year and the code of the ethics committee approbation. Especially for different countries that are involved.
Response 1: We thank the reviewer for pointing this out. The ethics committee approvals, including codes and approval dates for all countries involved in the study, are reported in the Institutional Review Board Statement section of the manuscript (Lines 648–671). For the reviewer’s convenience, we reproduce the relevant text below:
"The study was conducted in accordance with the Declaration of Helsinki and complied with all relevant ethical regulations.
Plasma samples were collected with appropriate ethical approvals as follows:
Spain: Approval was granted by the Institutional Review Boards of Hospital Universitario Vir-gen del Rocío and Hospital Universitario Virgen Macarena (Approval Code: CBG-CCOC-2018, dated July 3, 2018). The following institutions adhered to this central approval:
- Clinical Research Ethics Committee (CEIC), Hospital Clínico Universitario Lozano Blesa
- Clinical Research Ethics Committee with Medicines (CEIm), Parc de Salut Mar
- Fundación Asistencial Mutua Terrassa / Hospital Universitario Príncipe de Asturias
- Hospital Universitario Virgen de Valme
Additional independent approvals were obtained from:
- Complejo Hospitalario de Torrecárdenas (Internal Code: 29/2018, approved on September 26, 2018)
- Complejo Hospitalario de Navarra (Internal Code: 2018/60, approved on October 26, 2018)
- Hospital Universitari Germans Trias i Pujol (Internal Code: 18/168, approved on July 26, 2018)
Ukraine:
- Ethics Commission of Kyiv City Clinical Oncology Centre (Approval Code: 20001501/1, Identifi-cation No.: 13697965, approved on April 4, 2018)
- Local Ethics Commission of the Communal Institution, Kyiv Regional Council (Approval Code: 20001501/14, approved on January 31, 2018)
Germany:
- Ethics Committee of the Hamburg Medical Association (Ethik-Kommission der Ärztekammer Hamburg), Approval Code: PV5844, approved on November 4, 2018
Frozen tissue samples were obtained from the Indivumed Biobank repository, in compliance with IRB-approved protocols.”
Comment 2: In the materials and methods section, could the authors provide a summarized description of their patients. Perhaps as a one table for blood samples and another table for tissue samples, since the groups are different and state known data (like age +SD, sex, histopathological diagnosis, or any other relevant data).
Response 2: We appreciate this helpful suggestion. While a supplementary table summarizing the patient characteristics was included in the original version, we have now revised the manuscript to improve clarity. Specifically, we have added two separate tables: one for tissue samples (Table 1) and one for plasma samples (Table 2), each reporting age (mean ± SD), sex, and histopathological diagnosis. These tables are now clearly referenced in the revised Materials and Methods section under “4.1 Sample Preparation and Whole-Genome EM-seq” (Lines 445–476).
Comment 3: The authors have blood samples from different countries (Spain, Germany, and Ukraine). Did they check is there any significant difference between these groups of samples since the culture, geographical region and habits could influence different parameters like methylation.
Response 3: We thank the reviewer for this important point. Our cfDNA cohort was designed to minimize variability by applying consistent inclusion criteria across all centers. To assess potential confounding due to geographical origin, we performed a preliminary comparison of cfDNA methylation profiles between countries. A batch-wise ANOVA revealed no statistically significant differences (p > 0.05), indicating that methylation patterns were not meaningfully affected by site-specific or cultural factors. We have now included this clarification in the revised Materials and Methods section under “4.2 Sequence QC and DNA Methylation Analysis” (Lines 515–519).
Comment 4: Since the idea of this work was to help early detection of CRC, could the authors explain how do they envision using their results in clinical sense? Did they plan to use patterns and compare them? How could they be sure that specific pattern is typical for CRC and not some other disease of condition, since they don’t have any other comparison?
Response 4: We appreciate the reviewer raising this key translational question. Our study aims to define an early methylation signature that distinguishes low- and high-grade adenomas and persists across CRC progression. This biology-guided signature could serve as the foundation for noninvasive detection of high-risk lesions via cfDNA profiling.
We acknowledge, however, that specificity for CRC cannot be fully established without comparison to other diseases. Future work will expand the cfDNA cohort to include samples from patients with inflammatory bowel disease, other gastrointestinal cancers, and benign conditions to rigorously assess diagnostic specificity.
This limitation and clinical perspective have been addressed in the revised Discussion under subsection “3.6 Translational Outlook and Study Limitations” (Lines 424–442).
Comment 5: The discussion section is written as a mix of results with conclusions. The authors don’t discuss almost any of the works by other authors. Could the authors include more discussions into this section. Perhaps citing other works that have the same results supporting their own or other works that have opposite results and explaining why that could be.
Response 5: Thank you for this valuable feedback. We have revised the Discussion section to better separate the interpretation of our findings from the results and to enhance the scholarly context. The revised version includes an expanded review of relevant literature, including studies on cfDNA methylation, multi-omics classification strategies, and transcriptional subtyping of colorectal cancer. We also compare our biology-driven strategy to data-driven approaches reported in the literature, highlighting both similarities and key differences. These revisions help to frame our findings within the broader scientific landscape and underscore the novelty of our early-lesion anchored approach.
Reviewer 2 Report
Comments and Suggestions for Authors
This study focused on a comprehensive analysis of methylation signatures in dysplasia. Authors have utilized multi-model data and performed several analyses to understand the clusters and their biological relevance. This data-driven approach is interesting to characterize samples in a group manner. In general, the manuscript is written well with providing details in the context. However, I have the following suggestions.
1) Introduction: Authors have provided a nice context of CRC, more on biological insights, and the importance of methylation signatures to understand LGD and HGD. However; the story lacks how multi omics integration in this context is important or can be more powerful to characterize dysplasia progression. There are several studies focused on multiomics integration in cancer to understand subtypes, survival of patients, cancer diagnosis and progression, etc. Because the current study focused on multiomics integration, I would recommend adding a paragraph describing how people have used different approach to integrate multiomics data for predicting subtype or biomarkers. For example, PINSPlus (https://pubmed.ncbi.nlm.nih.gov/30590381/), CIMLR (https://pubmed.ncbi.nlm.nih.gov/30367051/), iCluF (https://pubmed.ncbi.nlm.nih.gov/38698887/), Subtype-GAIN (https://pubmed.ncbi.nlm.nih.gov/33599254/), etc. used different approaches for integrating multiomics data for specific biological objectives in cancer including CRC cancer (in some cases). This para will provide methodological and foundation support to the current study.
2) Have you evaluated Unsupervised hierarchical clustering using method such as Sihoutte score ? I think it is imortant to understand the heterogeneity of these clusters.
3) Line 145-146: The author mentioned that "Consistent with its poor survival outcomes, CL4 also exhibited the highest mutation burden, reinforcing the link between epigenetic dysregulation, genomic instability, and tumor aggressiveness". What about cluster 2 and 3 and their corresponding survival rates. Are they following the same pattern? Interpret this and use it for making the conclusions more strong.
Author Response
We greatly appreciate your thorough review of our manuscript, and the constructive feedback provided. Your comments have been instrumental in strengthening the presentation and scientific rigor of our study. Detailed responses to all your suggestions and corrections are provided below. Please feel free to reach out if further clarification is needed.
Comment 1: Introduction: Authors have provided a nice context of CRC, more on biological insights, and the importance of methylation signatures to understand LGD and HGD. However, the story lacks how multi omics integration in this context is important or can be more powerful to characterize dysplasia progression. There are several studies focused on multiomics integration in cancer to understand subtypes, survival of patients, cancer diagnosis and progression, etc. Because the current study focused on multiomics integration, I would recommend adding a paragraph describing how people have used different approach to integrate multiomics data for predicting subtype or biomarkers.
For example, PINSPlus (https://pubmed.ncbi.nlm.nih.gov/30590381/), CIMLR (https://pubmed.ncbi.nlm.nih.gov/30367051/), iCluF (https://pubmed.ncbi.nlm.nih.gov/38698887/), Subtype-GAIN (https://pubmed.ncbi.nlm.nih.gov/33599254/), etc. used different approaches for integrating multiomics data for specific biological objectives in cancer including CRC cancer (in some cases). This will provide methodological and foundation support to the current study.
Response 1: We thank the reviewer for this excellent suggestion. In response, we have added a dedicated paragraph to the Introduction (Lines 82–94) discussing the importance and increasing role of multi-omics integration in cancer research. We now reference several representative computational approaches (PINSPlus, CIMLR, iCluF, and Subtype-GAIN) which have been applied to cancer subtype classification, biomarker discovery, and clinical outcome prediction across various cancer types, including colorectal cancer. These examples provide a methodological foundation for our study and highlight the relevance of integrative analyses. Furthermore, we have revised the Discussion section to compare our biologically driven framework with these data-driven models. While many integrative approaches begin with unsupervised clustering of large public datasets, our strategy is grounded in histologically defined early lesions (LGD vs. HGD). We then validate this biology-guided methylation signature in cfDNA and interpret it within the TCGA multi-omics context, bridging early events with advanced disease subtypes. This comparison emphasizes the complementary strengths of our hypothesis-guided framework.
Comment 2: Have you evaluated Unsupervised hierarchical clustering using method such as Silhouette score? I think it is important to understand the heterogeneity of these clusters. 
Response 2: We appreciate this important methodological comment. While we did not apply the Silhouette score specifically, we performed unsupervised clustering using the ConsensusClusterPlus R package. This method provides a robust evaluation of cluster number and stability by applying subsampling-based consensus clustering, which mitigates sensitivity to noise and high dimensionality, common challenges in multi-omics datasets. Unlike single-metric approaches like the Silhouette score, ConsensusClusterPlus quantifies clustering stability across multiple iterations, making it a widely accepted approach for high-throughput data analysis. We have expanded the Methods section accordingly to describe this in more detail (Lines 542–550).
Comment 3: Line 145-146: The author mentioned that "Consistent with its poor survival outcomes, CL4 also exhibited the highest mutation burden, reinforcing the link between epigenetic dysregulation, genomic instability, and tumor aggressiveness". What about cluster 2 and 3 and their corresponding survival rates. Are they following the same pattern? Interpret this and use it for making the conclusions stronger.
Response 3: We thank the reviewer for this insightful comment. We have now extended our analysis and interpretation of clusters CL2 and CL3 to better contextualize their survival profiles and biological relevance. In the revised manuscript, we added a new figure panel (Figure 4b) showing mean methylation levels across all clusters (CL1–CL4), revealing a progressive gradient from low-risk to high-risk phenotypes. CL2 and CL3 occupy intermediate methylation states but differ significantly in transcription factor activity, EMT signaling, and survival outcomes. CL2 is associated with immune-modulatory features and relatively favorable prognosis, while CL3 shows enrichment in EMT-associated transcription factors and exhibits a survival profile closer to CL4, despite similar mutation burden to CL2. These results suggest a branched trajectory of tumor evolution rather than a linear progression, and further support the link between methylation burden, regulatory remodeling, and clinical outcome. These points have been fully incorporated into the restructured Discussion section.
We are again grateful for your constructive review, which has significantly improved the manuscript. We hope the revisions address your concerns satisfactorily.
Round 2
Reviewer 1 Report
Comments and Suggestions for Authors
I thank the authors for answering all my concerns. I think the manuscript has greatly improved and is now ready to be published.
Reviewer 2 Report
Comments and Suggestions for Authors
Authors have addressed all my concerns.